# Effectiveness and Durability of Transfer Training in Fencing

**DOI:** 10.3390/ijerph17030849

**Published:** 2020-01-29

**Authors:** Mateusz Witkowski, Łukasz Bojkowski, Krzysztof Karpowicz, Mariusz Konieczny, Michał Bronikowski, Maciej Tomczak

**Affiliations:** 1Adam Mickiewicz University in Poznań, Zagajnikowa 9, 60-568 Poznań, Poland; 2Poznan University of Physical Education, Królowej Jadwigi 27/39, 61-871 Poznań, Poland; 3Opole University of Technology, Prószkowska 76, 45-758 Opole, Poland

**Keywords:** bilateral transfer, fencing training, intermanual transfer, interhemispheric transfer

## Abstract

This paper reports the results of an experiment that aimed to study transfer training in fencing. Fencers from the experimental group underwent six-week transfer training while those from the control group underwent regular fencing training. The fencers’ performance was analyzed thrice: before the experimental training (pretest), immediately after it (posttest), and four weeks after it (retention test). Using a device that simulates fencing moves and analyzes the accuracy of such performance, participants completed, with both hands, three tests related to straight thrust accuracy. While no differences in hand grip strength was observed between the two groups across the three tests, significant differences occurred in terms of their performance on the device. The groups did not differ in the pretests and the retention tests. However, the fencers from the experimental group generally performed better in postests than prestests. These results show that bilateral transfer can be effective in foil fencing training, although its positive effects are short-term. In order to be effective, transfer training should be used as a regular training tool.

## 1. Introduction

Training methods based on bilateral transfer are mainly used in unilateral (asymmetric) sports [1,2]. Ruddy and Carson [3] demonstrated that physical training with one arm improves performance with the opposite (untrained) arm. This phenomenon is called intermanual transfer [3]. Similar effects have also been shown in many other motor tasks [4,5,6,7,8,9,10,11,12].

Intermanual transfer was suggested to result from plastic changes in the brain, which occur when new specific neural networks that control the physically trained effector are formed [13,14,15]. Interhemispheric interactions are often called interlimb interactions [10,12], intermanual interactions [16,17], contralateral interactions [18,19], and bilateral transfer [20]; henceforth, we will use the term bilateral transfer. Various aspects of motor control are asymmetrically distributed in both hemispheres of the brain [21,22]. The left hemisphere [23] is responsible for subtle and synchronous movements that require sequential and dynamic motor control while the right hemisphere for complex and intuitive movements, thus specializing in visual and spatial motion control [24].

The above relationships agree with the dynamic dominance hypothesis of handedness by Sainburg [25]. It assumes that the dominant system specializes in controlling dynamic traits while the non-dominant one controls the spatial characteristics of this movement. The structures connecting the hemispheres—above all, the corpus callosum, also referred to as the callosal commissure—affect the generation and effectiveness of individual human movements. The corpus callosum is the brain’s largest white tissue structure, responsible for interhemispheric relations [26]. Brain imaging research suggests links between the parietal and frontal lobes of the brain’s left hemisphere and the representation of skills related to the use of tools and other objects (transitive actions) [27].

To understand why training one limb brings about the enhancements in the performance of the opposite (untrained) limb, a cross-education phenomenon can help. Ruddy and Carson [3] showed that the unilateral execution of a movement task gave rise to a bilateral increase in corticospinal excitability. During unilateral practice, such distributed activity (the cross activation) leads to simultaneous adaptations in neural circuits that project to the muscles of the untrained limb, thus facilitating subsequent performance of the task. As Ruddy and Carson [3] explained, “alternatively, bilateral access models entail that motor engrams formed during unilateral practice, may subsequently be utilized bilaterally—that is, by the neural circuitry that constitutes the control centers for movements of both limbs.”

Motor learning in sports can be based on transfer training proposed by Starosta [28]. It uses a two-way flow of nerve impulses between the hemispheres. Theories about skill transfer appeared much earlier, however. Examples are Thorndike’s transfer of practice [29] and other theories about the transfer and generalization of learned motor skills to untrained skills, as well as Schmidt’s schema theory of discrete motor skill learning [30]. This schema assumes that each learned motor skill develops the whole schema, not just one particular relationship between a parameter and its result. Fencing is an asymmetric sport—the weapon is held in one hand only. Hence fencers are likely to have functional asymmetry of the body, which can affect their fencing performance [31,32,33]. In order to improve performance, thus this asymmetry might be taken into account in training. On the one hand, using the non-dominant arm affects neither reaction time (RT) nor choice reaction time (CRT) [34]. On the other hand, tests for the non-dominant arm preceded by tests for the dominant arm resulted in the reduction of RT and CRT, a process likely based on sensory priming mechanisms. According to the motor program theory of Schmidt and Wrisberg [35], well-learned motor habits coded in the motor areas of the cerebral cortex can be effectively performed on both right and left sides. This phenomenon should be considered in connection with the synaptic plasticity of the central nervous system, which allows one to effectively learn new movement forms and remove bad motor habits. Introduced into training by means of proper transfer training, the phenomenon might thus help improve fencing techniques and performance. It would do so by affecting the quality of new motor habits or improving those already acquired. In addition, opposing exercises increase the strength and duration of neural networks, thereby promoting better motor coordination by decreasing the bioelectric tension of the activated muscles [36].

Stimulating the interhemispheric connections by performing physical activities with the non-dominant arm, transfer training helps decrease the degree of an athlete’s asymmetry. Such training programs have been shown to improve the motor coordination and skills of athletes of various sports [37,38,39,40]. Witkowski et al. [41] showed that transfer training improved the accuracy of fencing actions in young fencers (below the age of 14 years). The authors studied only short-term effects, immediately after completing a six-week transfer training.

To the best of our knowledge, this has been the only study on transfer training in fencing. Inspired by its results, the present research studies transfer training in older foilists (age 14–20 years), in terms of their accuracy of straight thrust, the main action in foil fencing. Unlike Witkowski et al.’s [41], this study will analyze both short-term (i.e., immediately after completing the training) and longer-term (four weeks later) effects of transfer training in foil fencing. This work thus aims to study the effectiveness and durability of transfer training in foil fencing.

## 2. Materials and Methods

### 2.1. Participants

The group studied consisted of thirty-two foilists aged 14–20, both men (*n* = 16) and women (*n* = 16), all being at least intermediate-level fencers. The participants were randomly divided into two groups: (A) experimental (*n* = 16) and (B) control (*n* = 16). The randomization was balanced for sex and performance level: Four participants from each cohort group (i.e., junior women, cadet women, junior men, and cadet men) were randomly assigned to the experimental and control groups. So, both the experimental group and the control group included four cadet women, four junior women, four cadet men, and four cadet men, giving 16 participants per group and 32 participants altogether. The participants were chosen so as to represent similar fencing skills, so neither age nor category should have affected the results. All participants were healthy and post-puberty.

The experimental group underwent a six-week specialized transfer training program (described below) while the control group underwent traditional fencing training. To assess the effectiveness of the transfer training, straight thrust—a basic fencing technique—was used with three variants. The straight thrust skill had been mastered by all participants.

All the participants declared right-handedness during trainings and bouts, which was confirmed with the Edinburgh Questionnaire [42] and its modification [43].

The study was conducted in accordance with the ethical standards of the Declaration of Helsinki (Ethical Principles for Research Involving Human Subjects). All the participants were informed about the purpose of the study and agreed to participate on a voluntary basis. The study was approved by the Research Ethics Committee of the Karol Marcinkowski University of Medical Science (approval number 982/17).

### 2.2. Intervention

The experiment aimed to compare the accuracy of hits before and after the specialized training and to assess the durability of its effects [44]. For each participant, hit accuracy and handgrip strength were assessed at each test session (Figure 1).

The specialized training program was implemented in the experimental group for six weeks, 30 min a day. It started off with two weeks of whole-body coordination activities involving both sides of the body; the activities used regular size balls of various textures and weights, a small size tennis ball, and a floor ladder. Each exercise was practiced three times on the non-dominant side and once on the dominant side. These activities were followed by two weeks of eye-to-hand and eye-to-foot coordination exercises with the additional equipment: fencing foils (appropriate for the non-dominant hand) and the Favero EFT-1 electronic fencing target (Favero Eletronics Srl, Arcade, Treviso, Italy). During the final two weeks, the fencers practiced lunges, parries, and other fencing techniques with the non-dominant side, with proper footwork, and repeated the activities practiced earlier, again for both sides with the three-to-one ratio. More details of the program can be found in Witkowski et al. [41].

### 2.3. Measurements

#### 2.3.1. Hand Grip Strength Test

Grip strength of each hand was measured with a hand dynamometer (TAKEI 5401, P & A Medical Ltd. Chorley, UK). Each participant made two contractions per hand, changing the hand in the subsequent trials. The wrist was an extension of the forearm and continued from it in a straight line. During the test, the tested hand could not touch any other part of the body. A better measurement was used in the analyses. Hand strength was measured with a precision of up to 0.1 kg.

#### 2.3.2. Measuring the Accuracy of Hits

The accuracy of hits was assessed using the Favero EFT-1 electronic fencing target (FAVERO ELECTRONICS Srl, Arcade (Treviso), Italy; Figure 2). The device has five targets, each equipped with a LED light. Two targets are placed at a height of around 90 cm above the ground, and the other two at around 130 cm above the ground. The targets are spaced 30 cm apart. The fifth target is in the center of the device, at the intersection of the diagonals. The accuracy of hits and time of completing the test were measured using three different tests protocols. In each test, the participants aimed to hit red targets (which randomly lit up) with the tip of the foil (the button). This approach made it possible to precisely determine the accuracy of straight thrust in each protocol.

The test protocols are described below:(a)In test 5, a fencer hits two randomly occurring targets that light up in ten cycles. A cycle consists of two hits: the first target lights up, and when it is hit, a second one lights. The fencer aims to complete the ten cycles in the shortest time. The test measures the accuracy of the hits and the time of completing each cycle. For the analyses, the mean time for the ten cycles was used, with a precision of 0.01 s. At the beginning of the task, the fencers took the en garde position at a distance suitable for a straight thrust. The tests were performed with both the dominant and the non-dominant hand.(b)In test 7, the fencer hits three targets that light up one by one in ten cycles. Again, the fencer aims to complete ten cycles in the shortest time.(c)In test 9, the fencer hits three targets that light up simultaneously, in three cycles. The test enables one to measure the accuracy of hitting three targets in the shortest possible time, a task requiring the fencer to take a proper perception strategy.

Similar methodology, based on measuring the accuracy of hits, has been used in other studies [33,41,45].

### 2.4. Statistical Analysis

Two-way analysis of variance (ANOVA) was used, with one between-subject factor, called *group* (the experimental and the control groups), and one within-subject factor, called *phase* (repeated measurements in the three periods: before the experiment—the pretest; right after the experiment—the posttest; and four weeks after the experiment—the retest). The group-by-phase interaction was analyzed. If the interaction was significant for a particular variable, the Bonferroni test was used for post-hoc analyses. When the sphericity assumption was violated, the Greenhouse-Geisser correction was applied. Statistica 13.1 software (Texas, TX, USA) was used for the analyses.

## 3. Results

### 3.1. Testing the Accuracy of Hits

The group-by-phase interaction effect was significant for the dominant (right) hand for tests 5 and 7 (Table 1). The fencers from the experimental group needed a shorter mean time to complete the posttest than the pretest, and a longer mean time to complete the retest than the posttest. They completed the pretest and the retest in similar mean times. For test 7, the fencers from the experimental group completed their posttest in a shorter mean time than those from the control group. For tests 5, 7, and 9, the group-by-phase interaction effect was also significant for the non-dominant (left) hand (Table 1). The fencers from the experimental group needed a shorter mean time to complete the posttest than the pretest. A mean completion time of the retest was longer than that of the posttest. Mean times of completing the pretest and the retest; however, did not differ. The fencers in the experimental ground needed a shorter mean time to complete the posttest than the fencers in the control group (Table 1). The fencers in the control group usually completed their pretests, posttests, and retests in similar mean times (*p* > 0.05), except for test 5 performed with the left hand, in which a mean completion time was longer in the retest than that in the posttest.

### 3.2. Hand Grip Strength Test

The group-by-phase interaction was statistically non-significant for both right hand and left hand (Table 2).

## 4. Discussion

Despite clear evidence that bilateral transfer is effective in sports training, transfer training is rare among fencers. It can be useful not only in daily training, but also in training injured fencers. For example, a fencer with an injury of the dominant arm can train with the non-dominant one, thereby helping to maintain his or her special fitness. Moreover, when implemented in the early stages of training, transfer training can help increase fencers’ versatility.

The results of hit accuracy testing performed with right (dominant) hands suggest a positive effect of the transfer training program used in the experiment. However, a month later the effect was undetectable, indicating the effect’s short-term duration. To explain this, we may use a cross-education phenomenon. Lee and Carroll [45] found that unilateral resistance training gives the contralateral effects, which can be due to two possible reasons: the neural adaptation and long-lasting modification of motor pathways that project to the opposite untrained limb, and induced adaptations in motor areas that are primarily involved in the control of movements of the trained limb. The opposite (untrained) limb may access these modified neural circuits during maximal voluntary contractions in ways that are analogous to motor learning. Maybe this was missing in the experimental program, which focused on coordination skills.

The processes of teaching and generalizing motor skills in fencing are undoubtedly conditioned by other factors, such as age, training experience, and level of motor skills like coordination or special orientation [46]. Fencing’s unilateral nature also matters. The results of the present experiment suggested, however, that bilateral transfer—often marginalized and undervalued in asymmetric sports—can be an efficient tool in fencing training and that the speed and smoothness of motion can be efficiently transferred, corroborating Pan and van Gemmert’s [47] results of a visual and motor study describing the “dynamic dominance model”. Exercising the non-dominant limb can help shape the special abilities of athletes, likely enriching training and improving their performance.

Training the non-dominant hand in a continuous and systematic way can support the overall training of fencers in different age categories. The present research was planned based on the results of an earlier experiment in which transfer training had positive effects on fencers aged under 14 [41]. Starosta [28] claims that including symmetrical training supports improving asymmetric movements, thereby helping achieve better results in one-side dominant sports. Such an approach helps align both sides of the body; the greater the asymmetry of movements a discipline requires, the more intense this process should be. This idea was reflected in the experiment described here. We realized that symmetrization begins after mastering the activities with both hands, which is impossible in professional sports and does not guarantee effectiveness; therefore, we emphasized the importance of asymmetry in sport.

As follows from the above-mentioned literature and the results of the experiment on young fencers by Witkowski et al. [41], transfer between the dominant and non-dominant limbs can be aided by sensory feedback. An example is training by passive observation—one that is based on the idea of observational learning; during such passive observation, visual data are not controlled by the subject [48].

Other authors also confirm that symmetrical exercises positively affect the speed of learning in most—if not all—asymmetric sports and that this effect is accompanied by the feeling of relief and the decreasing of nervous tension. Wolf-Cvitak and Starosta [49] believe that thanks to bilateral transfer, symmetrical exercises help relax and develop motor coordination by forcing the second hemisphere to become more active. Their research shows that bilateral transfer from the left to the right hand has a greater effect on a learned motor function than from the right to the left hand.

In this study, the positive effects of the transfer training were short-term: Four weeks after the experimental training, they were detected for neither hand. This implies that transfer training needs to be systematic and its ad hoc use does not provide satisfactory long-term effects. Therefore, the systematic symmetrization of training will likely produce stable effects on the precision of the sensory-motor habits being learned. It should be emphasized that transfer training is especially efficient in asymmetric sports.

The retention tests in this study were not performed until four weeks after completing the experimental training. Thus the benefits may have decayed sooner than four weeks and at individually different rates. The accuracy of straight thrust was used to represent fencing performance and it was acknowledged that this may not represent a competitive fencing function.

In order to provide a more in-depth picture of the use of transfer training in foil fencing training, future research should deal with various aspects of the phenomenon. First, for the moment we have no knowledge of how advanced foilists would react to transfer training. Secondly, the retention tests in this study were performed four weeks after completing the experimental training, but its effects did not last so long. It does not mean, however, that they vanish right after finishing the training—their durability is worth checking in future experiments. Thirdly, other criteria than hand grip strength and the accuracy of straight thrust might be used to assess the efficiency of transfer training. Last but not least, transfer training itself can take various forms. So, designing effective transfer training programs constitutes yet another crucial topic for in-depth research.

## 5. Conclusions

The six-week transfer training of the non-dominant side of fencers significantly increased the accuracy of hits and the speed of movements with the dominant arm. The results, along with those reported by Witkowski et al. [41], demonstrated that bilateral transfer can be effective in foil fencing training. Its effects, however, are impermanent: They disappear some time after ceasing this type of stimulation. Thus, to maintain them for a longer time, transfer training should be applied systematically.

Quite likely most of the knowledge we have gained in this study can be extrapolated to other fencing disciplines and asymmetric combat sports, but this needs to be checked experimentally. Bilateral transfer is a complex phenomenon influenced by various processes, such as lateralization, symmetrization, and motor learning. Thus, we need to learn how the human brain functions, particularly under and after stimulation by various training techniques.

## Figures and Tables

**Figure 1 ijerph-17-00849-f001:**
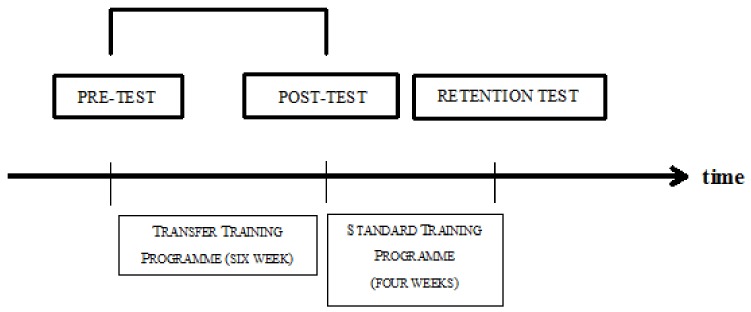
The timeline of the experiment.

**Figure 2 ijerph-17-00849-f002:**
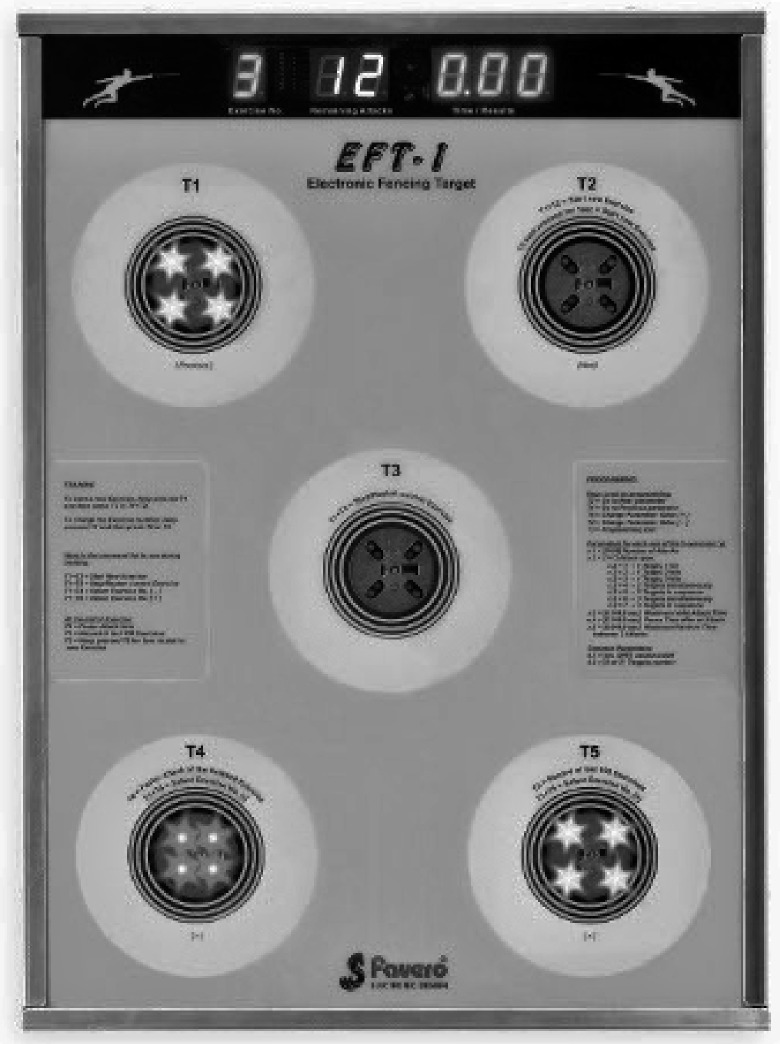
FAVERO: Electronic Fencing Target, EFT-1 (FAVERO ELECTRONICS Srl Arcade (TV)—Arcade, Italy).

**Table 1 ijerph-17-00849-t001:** Summary statistics for test completion time (in seconds) before the experiment (Pre), right after it (Post), and a month after it (Ret), for the right and left hand.

RH	**Group**	**Test 5**			**Test 7**			**Test 9**		
		**Pre**	**Post I**	**Ret**	**Pre**	**Post I**	**Ret**	**Pre**	**Post I**	**Ret**
		**M** **(SD)**	**M** **(SD)**	**M** **(SD)**	**M** **(SD)**	**M** **(SD)**	**M** **(SD)**	**M** **(SD)**	**M** **(SD)**	**M** **(SD)**
	EXP	1.93 ^1^ (0.22)	1.68 ^1,2^ (0.25)	2.02 ^2^ (0.18)	2.55 ^1^ (0.28)	2.14 ^1,2,3^ (0.36)	2.52 ^2^ (0.27)	2.44 (0.48)	2.21 (0.40)	2.45 (0.41)
	CON	1.91 (0.25)	1.94 (0.30)	2.04 (0.28)	2.56 (0.28)	2.54 ^3^ (0.39)	2.49 (0.29)	2.52 (0.33)	2.51 (0.44)	2.44 (0.37)
	Interaction	F (2.60) = 4.38, *p* < 0.05, η_p_^2^ = 0.13			F (1.25, 37.45) = 6.70, *p* < 0.01, η_p_^2^ = 0.18, ε = 0.62			F (1.34, 40.24) = 3.11, *p* > 0.05, ε = 0.67		
	Post-hoc comparisons #	1: *p* < 0.05; 2: *p* < 0.001			1: *p* < 0.001; 2: *p* < 0.01; 3: *p* < 0.01					
LH	**Group**	**Test 5**			**Test 7**			**Test 9**		
		**Pre**	**Post I**	**Ret**	**Pre**	**Post I**	**Ret**	**Pre**	**Post I**	**Ret**
		**M** **(SD)**	**M** **(SD)**	**M** **(SD)**	**M** **(SD)**	**M** **(SD)**	**M** **(SD)**	**M** **(SD)**	**M** **(SD)**	**M** **(SD)**
	EXP	2.28 ^1^ (0.20)	1.90 ^1,2,3^ (0.18)	2.26 ^2^ (0.20)	2.92 ^1^ (0.08)	2.45 ^1,2,3^ (0.27)	2.87 ^2^ (0.13)	2.87 ^1^ (0.19)	2.36 ^1,2,3^(0.29)	2.88 ^2^ (0.22)
	CON	2.32 (0.19)	2.20 ^3,4^ (0.18)	2.35 ^4^ (0.26)	2.92 (0.09)	2.90 ^3^ (0.12)	2.92 (0.12)	2.90 (0.20)	2.87 ^3^ (0.17)	2.86 (0.20)
	Interaction	F (2.60) = 10.07, *p* < 0.001, η_p_^2^ = 0.25			F (1.53,45.98) = 25.50, *p* < 0.001, η_p_^2^ = 0.46, ε = 0.77			F (1.18,35.43) = 20.33, *p* < 0.001, η_p_^2^ = 0.40, ε = 0.59		
	Post-hoc comparisons #	1: *p* < 0.001; 2: *p* < 0.001; 3: *p* < 0.01; 4: *p* < 0.05			1: *p* < 0.001; 2: *p* < 0.001; 3: *p* < 0.001			1: *p* < 0.001; 2: *p* < 0.001; 3: *p* < 0.001		

#—The same number for two means indicates that the difference between them is statistically significant, e.g., for test 5 performed with the right hand in the experimental group, the number 1 is put next to the mean values for Pre and Post I. This indicates that the difference between these two means (Pre and Post I) is statistically significant. RH—right hand; LH—left hand; EXP—experimental group; CON—control group; SD—standard deviation.

**Table 2 ijerph-17-00849-t002:** Summary statistics for the hand grip strength test (kg).

Group	Right Hand			Left Hand		
	Pre	Post I	Ret	Pre	Post I	Ret
	M (SD)	M (SD)	M (SD)	M (SD)	M (SD)	M (SD)
EXP	28.08 (9.54)	28.63 (9.49)	28.46 (9.36)	23.41 (6.70)	24.33 (6.52)	23.38 (6.23)
CON	27.09 (8.47)	26.98 (8.28)	26.94 (7.70)	24.28 (8.48)	24.36 (8.38)	24.14 (8.16)
Interaction	*F* (1.62,48.65) = 0.61; *p* > 0.05; *ε* = 0.81			*F* (1.49, 44.65) = 1.19; *p* > 0,05; *ε* = 0.74		

EXP—experimental group, CON—control group.

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
