# Peer review of "Effectiveness and Durability of Transfer Training in Fencing"

_ijerph, 2020, doi:10.3390/ijerph17030849_

Round 1

Reviewer 1 Report

Abstract:

The abstract is missing information about the statistical results and no mentions to the study conclusions.

Introduction:

- However, the research gap based on the existing body of literature is not clearly presented. What it was the novelty of the study?

Materials and Methods:

-The measurements are very weak for such sport as fencing, especially the accuracy-test that used.

- The authors didn't refer to any reference or study that modified the accuracy test       

Discussion:

- Since the discussion is not opened to the reader, it is difficult to see how your study moves this field forward and what is the contribution you aim to make to the theory. This can also be seen in the discussion section where the theoretical and practical implications are omitted.

- The discussion section does not help to identify what the results provided in the light of the existing body of knowledge that could bring the discussion forward. 

Conclusions:

- How could the findings be theoretically generalized and therefore also applied to other contexts?

Author Response

Dear Reviewer 1

Abstract:

The abstract is missing information about the statistical results and no mentions to the study conclusions.

We have revised the ending of the abstract, in order to add more information about the conclusions. You asked us to add statistical results to abstract in one of the previous versions, but, as before, we could not this for the following reasons. Abstract in this journal can only be 200-word long. In some cases, adding statistical results is easy—when the variable is easy to understand. But in other cases, including our manuscript, this is not so easy, because adding particular statistical results would call for much more detailed explanation of the parameters measured. We cannot simply provide statistical data without explaining the variables. We used a particular device in pretests, posttests and retention tests, and for each of them we had three device-specific tests (tests 5, 7, 9), each performed using both hands. To provide any quantitative data, we would have to explain all that (and more) so that the reader understands the data and the statistical results. But 200 words do not give sufficient space for doing so. This is why we described the results without providing detailed data.

Introduction:

- However, the research gap based on the existing body of literature is not clearly presented. What it was the novelty of the study?

Response: The last (slightly revised) paragraph of Introduction shows this novelty:

„To the best of our knowledge, this has been the only study on transfer training in fencing. Inspired by its results, the present research studies transfer training in older foilists (age 14–20 years), in terms of their accuracy of straight thrust, the main action in foil fencing. Unlike Witkowski et al.’s [41], this study will analyze both short-term (i.e., immediately after completing the training) and longer-term (four weeks later) effects of transfer training in foil fencing. This work thus aims to study the effectiveness and durability of transfer training in foil fencing.”

As you see, in particular this research is the first to study the durability of transfer training in fencing.

Materials and Methods:

-The measurements are very weak for such sport as fencing, especially the accuracy-test that used.

- The authors didn't refer to any reference or study that modified the accuracy test       

Response:  We have added three articles in which similar methodology was used.

Discussion:

- Since the discussion is not opened to the reader, it is difficult to see how your study moves this field forward and what is the contribution you aim to make to the theory. This can also be seen in the discussion section where the theoretical and practical implications are omitted.

- The discussion section does not help to identify what the results provided in the light of the existing body of knowledge that could bring the discussion forward. 

Response: Throughout several revisions of this manuscript, we have added some texts to discussion in order to improve the text as you suggested, but the other reviewers asked us to remove most of this text. So, we hope the current version, in which we added a little of new information to discussion, will satisfy both you and the other reviewers.

Conclusions:

- How could the findings be theoretically generalized and therefore also applied to other contexts?

Response: We believe that the results can quite likely be generalized to other fencing disciplines, but this—as any such extrapolation—needs further research. Can we generalize the results to other asymmetric combat sports? We are not sure, but this also needs further research. What we believe is true is that the positive effects we observed in foil fencing do suggest it’s worth to study transfer training in other unilateral combat sports.

The very last (revised) paragraph of the conclusions deals with this topic: “Quite likely most of the knowledge we have gained in this study can be extrapolated to other fencing disciplines and asymmetric combat sports, but this needs to be checked experimentally. Bilateral transfer is a complex phenomenon influenced by various processes, such as lateralization, symmetrization, and motor learning. Thus, we need to learn how the human brain functions, particularly under and after stimulation by various training techniques.”

We would like to thank you for your constructive comments throughout the whole review process of this manuscript. We do believe—and hope that you agree—that the current version is much better than the previous ones, thanks to yours and the other reviewers’ comments.

Reviewer 2 Report

Thanks again for your patience with reviewer comments/suggestions - I think the paper reads well - minor suggestions are   l98 . randomization was balanced   l99 . each such a cohort group     So, both the 
 .101  experimental group and the control group included four cadet women, four junior women, four cadet 
 .102  men, and four cadet men, giving 16 participants per group and 32 participants altogether. 

I like Table 1 but had difficulty following your superscripts and what they meant - maybe check APA guidelines but this needs to be clear to readers. 

best wishes

Author Response

Dear Reviewer 2

Thanks again for your patience with reviewer comments/suggestions - I think the paper reads well - minor suggestions are   l98. randomization was balanced   l99 . each such a cohort group     So, both the  .101  experimental group and the control group included four cadet women, four junior women, four cadet  .102  men, and four cadet men, giving 16 participants per group and 32 participants altogether.

I like Table 1 but had difficulty following your superscripts and what they meant - maybe check APA guidelines but this needs to be clear to readers. 

Response: We have added the following explanation under Table 1: “The same number for two means indicates that the difference between them is statistically significant, e.g. for test 5 performed with the right hand in the experimental group, the number 1 is put next to the mean values for Pre and Post I. This indicates that the difference between these two means (Pre and Post I) is statistically significant.” We have also revised the text, which, indeed, have lost its English quality during the several revisions we have made.

We would like to thank you for all your comments and suggestions throughout the whole peer review process, which helped us improve the manuscript.

Round 2

Reviewer 1 Report

Thank you for resubmitting the paper and doing the all comments.

This manuscript is a resubmission of an earlier submission. The following is a list of the peer review reports and author responses from that submission.

Round 1

Reviewer 1 Report

The article presents originality, focusing on the state of the art related to the study objective. The abstract conforms to the content. As the introduction starts from the whole for the specific, ending with the objective of study.

The methodology is well designed and very descriptive, the sample is balanced although small. There was a division of the control group and the experimental group based on experience. However, throughout the work, it is possibly not very clear regarding the choice of groups under 17 or under 20. This may bring contradictory or inappropriate results regarding the maturation of the athletes, or not, we don’t know.

The problem design is well designed and its intervention is very clear. Statistical procedures are appropriate for the purpose of study. The assessments are well laid out and enlightening.

The results are presented according to the three predicted evaluations and are almost always the experimental group that presents better results than the control group.

Even in the Hand grip strength test, although there are no statistical differences, the values are always better in the outgoing group.

Regarding the discussion, the authors present several possible justifications based on the study results and the literature, where although not always achieved the expected results, the evaluations and the study design bring transfer to practice and this is very important.

Indeed, one of the objectives of this type of research is precisely an attempt to contribute to the development of something, in this case of use with both hands, laterality and its implication in sport.

At the end of the discussion the authors present the possible limitations of the study. So as a possible short duration of the program. This represents a range of possible experiences in the future and this may have been a starting point.

When we come to the conclusions, the authors move to a situation of restlessness and questioning, which is the best way to go or to draw in the near future, is it really this or is it a different one? Which leads readers or new researchers to take up this topic and develop in the best way.

Finally, when we move to the bibliography, it is quite extensive, current, only the reference literature is a little older. But just some basic reference.

Reviewer 2 Report

Abstract:

The abstract is missing information about the statistical results and no mentions to the study conclusions.

Introduction:

- However, the research gap based on the existing body of literature is not clearly presented. What it was the novelty of study?

- The introduction was very long of sentences and paragraphs and missing a clearance of the current purpose of the manuscript.

- The theoretical section should be able to explain the topic as well, to understand the phenomenon under study.

Materials and Methods:

- I did not understand well how the participants were selected? The sample characteristics are too large according to age and education.

- The authors stated that subjects were men and women which divided into 2 groups (experimental and control). Are men group experimental only or mixed with the woman?

-The measurements are very weak for such sport as fencing, especially the accuracy-test that used.

- The authors didn't refer to any reference or study that modified the accuracy test       

Results:

- However, the authors used the Bonferroni test, the results section is missing any information about the statistical processes regarding the differences between the 3 tests.  

Discussion:

- Since the discussion is not opened to the reader, it is difficult to see how your study moves this field forward and what is the contribution you aim to make to the theory. This can also be seen in the discussion section where the theoretical and practical implications are omitted.

-The authors need to know the discussion and position of paper accordingly against this existing body of knowledge.

- The discussion section does not help to identify what the results provided in the light of the existing body of knowledge that could bring the discussion forward. In Discussions, you need to highlight the correlation of your results with previous studies.

Conclusions:

- How could the findings be theoretically generalized and therefore also applied to other contexts?

Reviewer 3 Report

ijerph-604703
Effectiveness and durability of transfer training in fencing

Thanks for the opportunity to review this manuscript. I found the research and the written presentation interesting and I think worthy of eventual publication.

For publication, I believe the manuscript needs

1. careful proofreading and likely some assistance with English grammar. Have said that, some sections of the manuscript are meticulously written and very clear - others have numerous errors and are difficult to follow.

2. condensing - I felt that the review of literature attempted to do too much. I suggest that the review needs to provide a context for your research and establish a research rationale but does not need to educate every reader. By this I think you explained the principles of contralateral transfer really well but then went into more detail and explored mechanisms which were not going to be explored or necessarily used within the discussion - I felt this section would benefit from abbreviation.

3. condensing of methods - again I felt there was great explanation and detail - without knowing fencing I soon grasped what was being done, but then I felt you overexplained those methods - eg. lines 157 - 171 on page 5. The statistical analysis section could also possibly be abbreviated - I think readers understand how and 2 way ANOVA works.

4. condense results. Each category of results appears to be presented 3 ways - text, a table, and a figure - when one way (maybe two in some cases) is all that is required. This leads to too much repetition and disrupts the flow of your manuscript - I think be more concise - handgrip for example just needs a sentence as there were no changes!!

I thought your discussion and conclusions were good and more appropriate to the manuscript.

Best wishes for getting this nice project published.

Reviewer 4 Report

This study aims to evaluate the effectiveness of transfer training in fencing. This type of training has previously been used for rehabilitation purposes and investigated using simple tasks like pinch grip. So, applying it to a complex sports movement in healthy athletes is novel. This study provides an interesting approach to enhancing skill development in athletes and may have important implications for training in unilateral sports. The findings suggest that any benefit are short-lived and further investigation is required.

The background is sufficient and provides a rationale for the study. The methodological approach and statistical analyses are appropriate.

Please provide the ethics approval number.

Line 76: You say that various studies confirm that, in fencing the weapon is held in one hand only. This seems obvious. Perhaps you mean that once a player chooses which hand to hold the weapon in, they never change or even practice with the non-preferred hand. I don't see that a study needs to be conducted to confirm this. Please clarify.

Lines 121 - 126: This description is confusing. Greater detail of the accuracy tests is provided later so please add a sentence to explain that the tests are detailed in section 2.3.

Figure 1 is not mentioned in the main text. The information the figure provides should be to add clarity or support to the text. Also the figure does not make it clear what the difference between the groups is.

Statistical analyses are appropriate.

Line 203: Change "Like in test 5..." to As in test 5..."

This study provides a foundation for further research. It would be interesting to see if a longer intervention would have longer retention and to see if bilateral transfer would be similar in other sporting contexts.